Exogenous hydrogen sulfide improves salt stress tolerance of Reaumuria soongorica seedlings by regulating active oxygen metabolism

Liu Hanghang
Chong Peifang zhongpf@gsau.edu.cn
Liu Zehua
Bao Xinguang
Tan Bingbing
Gansu Agricultural University , Lanzhou , China
Irfan Mohammad
Electronic publication date: 2023 Aug 24
Publication date: 2023
Volume: 11
Electronic Location ID: e15881
Received 2023 Mar 31; Accepted 2023 Jul 20
Copyright: ©2023 Liu et al.
Copyright year: 2023
Copyright holder: Liu et al.
License: This is an open access article distributed under the terms of the Creative Commons Attribution License, which permits unrestricted use, distribution, reproduction and adaptation in any medium and for any purpose provided that it is properly attributed. For attribution, the original author(s), title, publication source (PeerJ) and either DOI or URL of the article must be cited.
License URL: https://creativecommons.org/licenses/by/4.0/

Keywords: Reaumuria soongorica, Salt stress, ROS, H2S

Funding: National Natural Foundation of China 32160407 Grass Industry Open Project KLGE202215 Regular Science and Technology Assistance to Developing Countries KY202002011 Gansu Provincial Science and Technology Program 23JRRA1451 22JR5RA836 This work was supported by the National Natural Foundation of China (No. 32160407); the Grass Industry Open Project (No. KLGE202215); the Regular Science and Technology Assistance to Developing Countries (No. KY202002011); the Outstanding Doctoral Projects Funded by Gansu Provincial Science and Technology Program (No. 23JRRA1451); the Outstanding Doctoral Projects Funded by Gansu Provincial Science and Technology Program (No. 22JR5RA836). The funders had no role in study design, data collection and analysis, decision to publish, or preparation of the manuscript.

==============================
Hydrogen sulfide (H2S), as an endogenous gas signaling molecule, plays an important role in plant growth regulation and resistance to abiotic stress. This study aims to investigate the mechanism of exogenous H2S on the growth and development of Reaumuria soongorica seedlings under salt stress and to determine the optimal concentration for foliar application. To investigate the regulatory effects of exogenous H2S (donor sodium hydrosulfide, NaHS) at concentrations ranging from 0 to 1 mM on reactive oxygen species (ROS), antioxidant system, and osmoregulation in R. soongorica seedlings under 300 mM NaCl stress. The growth of R. soongorica seedlings was inhibited by salt stress, which resulted in a decrease in the leaf relative water content (LRWC), specific leaf area (SLA), and soluble sugar content in leaves, elevated activity levels of superoxide dismutase (SOD), peroxidase (POD), and catalase (CAT); and accumulated superoxide anion (O2–), proline, malondialdehyde (MDA), and soluble protein content in leaves; and increased L-cysteine desulfhydrase (LCD) activity and endogenous H2S content. This indicated that a high level of ROS was produced in the leaves of R. soongorica seedlings and seriously affected the growth and development of R. soongorica seedlings. The exogenous application of different concentrations of NaHS reduced the content of O 2–, proline and MDA, increased the activity of antioxidant enzymes and the content of osmoregulators (soluble sugars and soluble proteins), while the LCD enzyme activity and the content of endogenous H2S were further increased with the continuous application of exogenous H2S. The inhibitory effects of salt stress on the growth rate of plant height and ground diameter, the LRWC, biomass, and SLA were effectively alleviated. A comprehensive analysis showed that the LRWC, POD, and proline could be used as the main indicators to evaluate the alleviating effect of exogenous H2S on R. soongorica seedlings under salt stress. The optimal concentration of exogenous H2S for R. soongorica seedlings under salt stress was 0.025 mM. This study provides an important theoretical foundation for understanding the salt tolerance mechanism of R. soongorica and for cultivating high-quality germplasm resources.

Introduction

Salt stress, one of the main environmental factors that limits plant growth and crop productivity, can deteriorate soil characteristics and quality, seriously affecting ecology and agricultural resource utilization (Turk et al., 2014). The problem of soil salinization is widespread throughout the world and is particularly significant in arid and semi-arid regions. In particular, the problems of land desertification and soil salinization have become increasingly severe in northwest China. The accumulation of soil salts has been further aggravated by the combination of drought and low rainfall, arid climate, high evaporation and unreasonable irrigation methods in the region (Jesus et al., 2015). Salt stress can induce changes in morphological, physiological and metabolic responses in plants, activating osmotic stress pathways and reactive oxygen species (ROS) mechanisms (Genisel, Erdal & Kizilkaya, 2015). In addition, ROS can interact with a variety of molecular mechanisms to trigger a range of stress responses, including membrane lipid peroxidation, protein denaturation, and inactivation of associated antioxidant enzymes (Li et al., 2012). Plants have evolved a wide array of non-enzymatic and enzymatic defense mechanisms to alleviate salt stress-induced ROS to scavenge free radical-induced damage, and to protect them from deleterious oxidative stress (Yadav et al., 2020). Antioxidant defense systems include non-enzymatic antioxidants such as proline, malondialdehyde, and soluble proteins, while antioxidant enzymes include superoxide dismutase (SOD), catalase (CAT), and peroxidases (POD). Studies have shown that the antioxidant system plays an important role in protecting plants from salt stress-induced oxidative damage (Apel & Hirt, 2004; Mittler et al., 2011). Therefore, how to effectively regulate the antioxidant enzyme activity and the level of non-enzymatic compounds in plants is necessary to improve salt tolerance in plants.

R. soongorica is a xerophytic and halophytic small perennial shrub of the Tamaricaceae family (Zhang et al., 2020). The species is primarily distributed in the northwestern region of China, and its exceptional stress resistance is very important for ecological protection and construction in desert and grassland areas (Chong et al., 2019). Its leaves are rich in proteins, fats, and trace elements. Additionally, it serves as the main vegetation for the construction of fodder bushes and the cultivation of degraded grasslands (Zhang et al., 2020). The unique salt gland structure of R. soongorica is an essential guarantee for its survival in saline environments. And the main function is that when plants secrete salt, the salt in the tissues around the salt glands is usually first transported to the large vesicles that have a collecting function. Subsequently, the secretory cells transport the accumulated substances into their small vesicles, which constantly migrate towards the side of the cell membrane and are ultimately excreted by the outer secretory cells through their plasma membrane (Semenova, Fomina & Biel, 2010). Meanwhile, R. soongorica has a high ecological niche width for different salinity gradients, but salinity is one of the main factors affecting the distribution of R. soongorica communities (Yan, Chong & Zhao, 2022). Studies have shown that R. soongorica is a typical salt-secreting plant, which can alleviate the effects of salt damage through leaf salt secretion and regulation of reactive oxygen species (ROS) under salt stress (He et al., 2019).

Hydrogen sulfide (H2S) is an inorganic compound like CO and NO (Goyal, Jhanghel & Mehrotra, 2021). It can be used as a signaling molecule to participate in the regulation of various physiological functions in plants, including promoting seed germination, regulating photosynthesis, root development and stomatal movement, and also involved in resistance to stresses such as heavy metals, salt, drought and high temperature (Lai et al., 2014). It has been shown that the main effect of H2S is to significantly improve the salt tolerance of plants by maintaining Na+/K+ homeost1asis (Huang, Huo & Liao, 2021). The application of exogenous H2S can effectively increase the antioxidant level of pea seedlings under salt stress (Singh et al., 2015), improve the tolerance of tomato seedlings to NO3− stress (Guo et al., 2018), and promote the growth of wheat (Deng et al., 2015). It plays a very important role in the response to abiotic stress, fruit ripening and improving fruit quality traits (Shi et al., 2015; Tayal, Kumar & Irfan, 2022; Zhang et al., 2022). Sodium hydrosulfide (NaHS) is a commonly used exogenous H2S donor in biological studies due to its ability to dissociate into Na+ and HS− in plant organisms. This allows the combination of H+ with HS− in the body to form H2S. This process establishes a dynamic balance to achieve exogenous H2S application (Hosoki, Matsuki & Kimura, 1997).

In recent years, as the use of small molecules to regulate the mechanism of plant salt tolerance has been more widely studied, H2S as a new type of small molecule in the regulation of plant physiological metabolism under adversity stress has increasingly become a hotspot. However, it is not known whether exogenous H2S treatment also induces salt stress tolerance in R. soongorica seedlings. Therefore, we proposed the hypothesis that the salt tolerance of R. soongorica seedlings could be improved by exogenous H2S treatment through modulation of antioxidant mechanisms. In this study, the effects of different concentrations of exogenous NaHS on osmotic substance content and oxidoreductase activity of R. soongorica seedlings under salt stress were investigated. To explore the mechanism by which exogenous NaHS mitigates the impairment of R. soongorica seedling growth under salt stress and to screen for the optimal exogenous NaHS concentration for ameliorating salt stress. It is important to further explore and study how to effectively improve the survival and reproduction rate of R. soongorica seedlings in saline habitats, and to cultivate high-quality germplasm resources of R. soongorica and apply them in ecological vegetation construction.

Materials & Methods

Plant material and culture

The seeds of Reaumuria soongorica were collected from Yangxiaba, Liangzhou District, Wuwei City, Gansu Province. After selecting the full-grain R. soongorica seeds, they were sterilized with 0.3% KMnO4 solution for 15 min and rinsed five times with deionized water. Subsequently, the surface water of the seeds was drained before sowing into pre-sterilized culture substrate consisting of peat soil, vermiculite and perlite mixed at 3:1:1 by volume; each pot was filled with approximately 2 kg. The seedlings were cultivated in a well-ventilated and light-permeable artificial shelter around the research base of the Gansu Agricultural University campus. When the seedlings grew to 10–12 cm under natural light, the uniformly sized and grown seedlings were selected for the experiment, and three plants were kept in each pot.

Experimental design

The experiment was designed using a completely randomized block design, and the salt stress concentration used in the experiment was a 300 mM NaCl solution that had been pre-screened by our team for its ability to effectively inhibit the growth of R. soongorica seedlings (Yan, Chong & Zhao, 2022). The salt stress treatment was recorded at the beginning of the experiment after the 50 mM NaCl solution was increased to the treatment concentration within 24 h to prevent salt shock. The exogenous donor of H2S, sodium sulfide (NaHS, Sigma), and nine treatments were established by foliar spraying. The control (CK) was watered with 1/2 Hoagland, the control (S) was watered with 1/2 Hoagland-prepared 300 mM NaCl solution, and both groups were foliar sprayed with distilled water. The other seven H2S treatments were watered with 1/2 Hoagland-prepared 300 mM NaCl solution and foliar sprayed with NaHS solutions at concentrations of 0.01, 0.025, 0.05, 0.1, 0.25, 0.5, and 1 mM, respectively. Three replicates of each treatment were used to determine physiological and plant morphological indices in two groups of 54 pots. The leaves of R. soongorica seedlings were sprayed with the corresponding concentration of NaHS solution every morning at 10 am until the leaves dripped, and all relevant indices were measured at 0, 7, 14, 21 and 28 days, respectively.

Determination of plant height, ground diameter, leaf relative water content, specific leaf area and biomass

Plant height by measuring the distance from the base of the plant to the top of the main stem with a straight edge. The ground diameter by measuring the smoothness of the plant 1 cm from the soil surface in the pot with a vernier caliper. And the average relative growth rate of plant height was calculated using the following formula: RGR = (lnW2 − lnW1)/(T2 − T1), where W1 is the plant height at T1 and W2 is the plant height at time T2. The absolute growth rate of ground diameter was calculated using the following equation: AGR = (W2 − W1)/(T2 − T1), where W1 is the ground diameter at time T1 and W2 is the ground diameter at time T2. To measure the leaf relative water content (LRWC) and specific leaf area (SLA), we separated the leaves from the plants at 10 am to avoid the effect of light on water loss from the separated leaves. The SLA was calculated using the following equation: SLA = L/W, where L is the total leaf area and W is the leaf dry weight. The LRWC was calculated using the following equation: LRWC = (FW − DW)/(TW − DW),  where FW is the fresh weight of the leaf, DW is the dry weight of the leaf (measured after drying the leaf at 105 °C for 24 h), and TW is the turgid weight, which was measured after keeping the leaves in deionized water for 24 h. The biomass of R. soongorica seedlings was measured after 28 d. R. soongorica seedlings were rinsed with deionized water and dehydrated. The seedlings were cut from the rootstock union, divided into above and below ground parts and weighed separately for fresh weight; then baked in an oven at 105 °C for 15 min and dried at 75 °C to a constant weight and weighed for dry weight.

Antioxidant enzyme activities determination

Briefly, leaf samples (0.1 g fresh weight, FW) were homogenized in 3 mL extraction buffer (5 mM EDTA, 0.5% PVP, 50 mM phosphate buffer, pH 7.8). The homogenate was centrifuged at 4 °C, 15,000 ×g for 15 min, and the resulting supernatant was used for the enzyme assay.

SOD activity was determined by the spectrophotometric method described by Qiu et al. (2014), which measures the inhibition of the photochemical reduction of nitroblue tetrazolium (NBT) at 560 nm. The reaction solution (3 mL) consisted of 0.1 mL enzyme extract (0.1 mL deionized water as control), 1.5 mL of 50 mM phosphate buffer (pH 7.8), 0.3 mL of 0.1 mM ethylenediaminetetraacetic acid (EDTA), 0.3 mL of 130 mM methionine, 0.3 mL of 0.75 mM NBT, 0.3 mL of 0.02 mM riboflavin and 0.5 mL of deionized water. The reaction was initiated by placing the tubes under natural light and terminated after 10 min. Non-illuminated and illuminated reactions without enzyme extract were used as calibration standards. One unit of SOD activity was defined as the amount of enzyme required to cause 50% inhibition of NBT reduction.

CAT activity was determined in a 3.0 mL reaction solution containing 0.1 mL enzyme extract, 25% H2O2 (0.1 µM EDTA, 100 mM phosphate buffer, pH 7.0) according to Qiu et al. (2011). The absorbance was declined at 240 nm for 2 min. One unit of CAT activity was defined as a change in absorbance of 0.01 for 1 min caused by the enzyme extract.

The POD activity was determined by measuring the oxidation of guaiacol with H2O2 according to the method described by Qiu et al. (2011). The enzyme extract (0.5 mL) was added to the reaction solution consisting of 1.8 mL guaiacol solution and 0.7 mL H2O2 solution. The addition of the enzyme extract initiated the reaction and the increase in absorbance was recorded at 470 nm for 2 min. Enzyme activity was quantified by defining a 0.01 increase in absorbance for 1 min as 1 viability unit.

Determination of osmotic regulation substances and indexes related to membrane peroxidation

The content was determined by the method described by Bao et al. (2020). Leaf samples (0.1 g fresh weight, FW) were homogenized in 6 mL of 3% sulfosalicylic acid followed by heating in boiling water for 10 min. After filtration, 1 mL of the extract (toluene as control) was heated with 2 mL glacial acetic acid and 2 mL acidic ninhydrin in boiling water for 15 min. After cooling to room temperature, 4 mL of toluene was added and the toluene phase was separated by resting for 3 h. The absorbance was measured at 520 nm, and a 2–30 mg/mL standard solution was used to draw the standard curve.

The soluble protein content was determined by the method described by Bao et al. (2020). Leaf samples (0.2 g fresh weight, FW) were homogenized with 5 mL extraction buffer (5 mM EDTA, 0.5% PVP, 50 mM phosphate buffer, pH 7.8) and centrifuged at 4 °C, 15,000 × g for 10 min. The supernatant (0.5 mL) was transferred to a test tube, then 0.5 mL deionized water and 5 mL Coomassie brilliant blue G-250 were added. The absorbance was measured at 595 nm after 2 min and the soluble protein content was determined from a standard curve.

The concentration of malondialdehyde (MDA) was determined using the method described by Qiu et al. (2014). Briefly, leaf samples (0.2 g fresh weight, FW) were homogenized with 5 mL of chilled 10% trichloroacetic acid (TCA) and centrifuged at 4 °C, 15,000 × g for 10 min. A total of 1 mL of the supernatant was mixed with 2 mL of 0.6% thiobarbituric acid (TBA) in boiling water for 15 min and then rapidly cooled in an ice-bath. The mixture was centrifuged again. The same procedure was performed with deionized water to obtain the control. The absorbance was measured at 532 nm, 600 nm and 450 nm.

The soluble sugar content was determined by the method described by Bao et al. (2020). Leaf samples (0.1 g fresh weight, FW) were placed in a 10 mL centrifuge tube, 6 mL of 80% ethanol solution was added and heated in boiling water at 80 °C for 30 min, then rapidly cooled to room temperature and centrifuged at 3,500 xg for 10 min. The supernatant was transferred to a volumetric flask and 6 mL of 80% ethanol was added to the precipitate. The extraction was repeated twice as described above. The supernatant was combined in a volumetric flask and the volume was fixed at 25 mL. A total of 1 mL of extract and 5 mL of anthraquinone-H2SO4 reagent were mixed and heated in boiling water for 10 min, then cooled to room temperature and the absorbance was measured at 620 nm and the soluble sugar content was determined using a standard curve.

For the determination of O2− in the leaf tissue of R. soongorica as described by Yan, Chong & Zhao (2022), the procedure described by was used with some modifications. Briefly, 0.5 g of fresh leaf samples were ground into a homogenate with 10 mL of 50 mM phosphate buffer (pH 7.8), followed by centrifugation at 15, 000 × g at 4 °C for 10 min. A total of 2 mL of the supernatant extract was added to 1.5 mL of 50 mM phosphate buffer and 0.05 mL of 10 mol hydroxylamine hydrochloride for 20 min. Then, 2 mL of the above reaction solution was withdrawn into a test tube and 2 mL of 17 mM p-aminobenzene sulfonic acid and 2 mL of 7 mM α -naphthylamine were added in a 30 °C water bath for 30 min. The absorbance of the reaction solution was measured at 530 nm.

Determination of endogenous hormones H2S and L-cysteine dehippuricase (LCD)

Endogenous H2S content was determined by the method described by Shi et al. (2015). Briefly, leaf samples (0.5 g fresh weight, FW) were homogenized in 7.0 mL extraction buffer (0.25% (w/v) zinc acetate, 10 mM EDTA, 100 mM phosphate buffer, pH 8.0) and centrifuged at 4 °C, 15,000 × g for 15 min. To the supernatant extract (3 mL), 1 mL of 5 mM N, N-dimethyl-p-phenylenediamine was added and shaken slowly for 12 min, then 1 mL of 150 mM ferric chloride (FeCl3) was added and allowed to fully react for 15 min. The absorbance of the reaction solution was measured at 667 nm.

L-cysteine desulfhydrase (LCD) activity was determined by the release of H2S from L-cysteine according to the method described by Jiang et al. (2019). Briefly, leaf samples (0.5 g fresh weight, FW) were homogenized in 7.0 mL extraction buffer (10 mM EDTA, 100 mM phosphate buffer, pH 7.4) and centrifuged at 4 °C, 15,000 × g for 15 min, the reaction was initiated by mixing 2.0 mL L-cysteine desulfhydrase reaction solution (2.5 mM dithiothreitol, 0.8 mM L-cysteine, 100 mM Tris–HCl, pH 8.0) with 1.0 mL LCD extraction solution. After 15 min, the reaction was terminated by adding 1 mL of 30 mM ferric chloride dissolved in 1.2 N HCl and 1 mL of 20 mM N, N-dimethyl-p-phenylenediamine dissolved in 7.2 N HCl. The absorbance of the reaction solution was measured at 670 nm.

Comprehensive evaluation of exogenous H2S to alleviate R. soongorica seedling leaves under salt stress

Principal component score

The principal component score was calculated using the following method: Fj=U1X1+U2X2+…+UjXjj=1,2,3,…,n

First, the arithmetic square root of the corresponding eigenvalues of the principal components (Table 1) is calculated. Then, the coefficient corresponding to each indicator (Uj) in the principal component is obtained by dividing the principal component loadings of each indicator (Table 2) by the arithmetic square root of the corresponding eigenvalues of the principal component. The original data of each indicator (Xj) are standardized and the standardized value is multiplied by (Uj), which is the score of each principal component (Fj).

Table 1 Explanation of the total variance of the principal components of R. soongorica at different treatment times.

Treatment time (day)	Principal component	Characteristic value	Contribution rate %	Cumulative contribution rate %	
7	1	5.896	42.115	42.115	
2	5.456	38.970	81.085	
3	2.042	14.588	95.674	
14	1	5.915	42.251	42.251	
2	5.535	39.539	81.790	
3	1.540	10.999	92.789	
21	1	6.572	46.945	46.945	
2	5.049	36.063	83.009	
3	1.371	9.790	92.799	
28	1	7.648	54.630	54.630	
2	4.302	30.730	85.361	
3	1.433	10.234	95.595	

Table 2 Principal component loading matrix of each indicator variable of R. soongorica at different treatment times.

T (day)	PCL	CAT	H2S	LCD	MDA	POD	SOD	O2−	SP	Pro	SS	GD-AGR	PH-RGR	LRWC	SLA	
7	1	0.533	0.308	0.391	−0.589	0.398	0.534	−0.525	0.457	−0.536	0.842	0.818	0.853	0.940	0.902	
2	0.784	0.480	0.424	0.751	0.911	0.842	0.834	0.813	0.835	0.053	−0.406	−0.226	0.095	−0.216	
3	−0.297	0.808	0.808	−0.110	−0.038	0.037	−0.142	−0.346	−0.012	−0.519	0.236	−0.293	0.244	−0.150	
14	1	0.788	0.708	0.528	0.677	0.915	0.734	0.779	0.627	0.779	0.330	−0.596	−0.729	0.039	−0.272	
2	0.485	0.450	0.432	−0.640	0.347	0.642	−0.566	0.659	−0.574	0.604	0.707	0.518	0.961	0.910	
3	0.322	−0.486	−0.701	−0.046	−0.006	−0.018	0.252	0.405	0.018	0.504	−0.269	0.382	−0.080	0	
21	1	0.866	0.875	0.852	−0.207	0.910	0.962	0.025	0.852	−0.367	0.603	−0.310	−0.207	0.880	0.627	
2	0.190	0.154	0.155	0.889	0.406	0.229	0.915	−0.095	0.902	−0.147	−0.902	−0.868	−0.409	−0.736	
3	0.420	−0.425	−0.486	−0.214	−0.023	−0.072	0.351	0.450	−0.028	0.519	0.226	−0.193	−0.190	−0.079	
28	1	0.846	0.867	0.845	−0.166	0.957	0.969	−0.227	0.715	−0.619	0.889	−0.292	−0.489	0.968	0.764	
2	0.196	0.169	0.142	0.942	0.238	0.206	0.857	0.104	0.754	−0.169	−0.929	−0.806	−0.194	−0.579	
3	0.388	−0.452	−0.507	−0.210	−0.051	−0.022	0.414	0.666	0.113	0.331	0.157	0.065	0.053	−0.069	
Notes.

The T in the table represents treatment time; PCL represents the principal component load. The letters above are CAT (catalase), H2S (endogenous H2S), LCD (L-cysteine desulfhydrase), MDA (malondialdehyde), POD (peroxidase), SOD (superoxide dismutase), O2− (superoxide anion), SP (soluble protein), Pro (proline), SS (soluble sugar), GD-AGR (absolute growth rate of ground diameter), PH-RGR (relative growth rate of plant height), LRWC (leaf relative water), SLA (specific leaf area), respectively. The same as following.

The composite score of the principal components is calculated using the following method: F=F1R1+F2R2+…+FjRjj=1,2,3,…,n

The sum of the product of each principal component score (Fj) and the corresponding cumulative contribution rate (Rj) (Table 1) is the composite principal component score (F).

Affiliation function method

The value of the affiliation function of each indicator is calculated using the following method: UXj=Xj−XminXmax−Xminj=1,2,3,…,n

where Xj is the value of j indicators; Xmin is the minimum value of j indicators; Xmax is the maximum value of j indicators.

Statistical analysis

All data were analyzed by one-way ANOVA using IBM SPSS 26.0 software, and mean differences were compared by Duncan’s multiple range test, and the means of all data were calculated using principal component analysis and the affiliation function. Plotting was performed by Origin 2021 software. The distributions of the values in all tables are shown as standard errors of the means.

Results

Exogenous H2S affects the growth of R. soongorica seedlings under salt stress

As shown in Fig. 1A, salt stress treatment (300 mM NaCl) effectively inhibited the growth rate of plant height of R. soongorica seedlings compared with the control plant (CK). The relative growth rate (RGR) of R. soongorica plant height under salt stress treatment decreased by 50.4%, 82.5%, 77.6%, and 73.6% with increasing time (7 d, 14 d, 21 d, and 28 d) compared with CK, respectively. Under the treatment of different concentrations of NaHS, compared with single salt stress (S) alone, the RGR of plant height first increased and then decreased with increasing NaHS concentration. Compared with S plants, the RGR of plant height increased significantly with increasing NaHS treatment time. It indicated that exogenous NaHS effectively alleviated the growth inhibition of R. soongorica seedlings by salt stress.

Figure 1 Effects of exogenous H2S (NaHS) on the relative growth rate (RGR) of plant height (A), absolute growth rate (AGR) of ground diameter (B), leaf relative water content (C) and specific leaf area (D) of R. soongorica seedlings.

As shown in Fig. 1B, the absolute growth rate (AGR) of ground diameter of R. soongorica under salt stress treatment was significantly decreased by 77.4%, 59.1%, 20.0%, and 56.0% compared with CK at 7, 14, 21, and 28 d, respectively, which significantly inhibited the growth of ground diameter. Compared with S plants, under the treatment of different concentrations of NaHS, the AGR of ground diameter first increased and then decreased with increasing NaHS concentration. Except for the most effective AGR of ground diameter of R. soongorica under 0.1 mM NaHS treatment at 7 d, the most significant growth rate of the AGR of ground diameter was observed at 14, 21, and 28 d under 0.05 mM NaHS treatment. It shows that exogenous NaHS effectively promoted the growth of R. soongorica ground diameter under salt stress.

It can be seen from Fig. 1C that the LRWC of R. soongorica under salt treatment gradually decreased with time compared with CK. It decreased by 3.8%, 3.7%, 2.7%, and 1.7% at 7, 14, 21, and 28 d, respectively. Under different concentrations of NaHS, the LRWC first increased and then decreased with increasing NaHS concentration. Compared with S plants, the LRWC gradually increased with increasing NaHS treatment time. Through treatments in which the most effective NaHS concentrations at each period were 7 d (0.05 mM NaHS), 14 d (0.1 mM NaHS), 21 d (0.1 mM NaHS), and 28 d (0.025 mM NaHS). The LRWC increased by 5.8%, 5.9%, 6.3%, and 5.2%, respectively, compared with S plants. The results showed that exogenous NaHS effectively inhibited the decrease in the LRWC of R. soongorica under salt stress.

As shown in Fig. 1D, the SLA of R. soongorica under salt treatment decreased continuously with time compared to CK. It decreased by 20.3%, 13.3%, 10.9%, and 12.6% at 7, 14, 21, and 28 d, respectively. After the application of different concentrations of NaHS, the SLA showed the trend of first increased and then decreased with increasing NaHS concentration. Compared with S plants, the SLA increased significantly with increasing NaHS treatment time. And the NaHS treatment concentration (0.05 mM) was the most significant at 7 and 21 d, the SLA increased by 24.0% and 14.4% compared with S plants, respectively; the NaHS treatment concentration (0.1 mM) was the most significant at 14 and 28 d, the SLA increased by 16.6% and 19.8% compared with S plants, respectively. This indicates that exogenous NaHS can effectively alleviate the effect of salt stress on the SLA of R. soongorica.

Exogenous H2S affects non-enzymatic antioxidant levels of R. soongorica seedlings under salt stress

It can be seen from Fig. 2A that the proline content in R. soongorica leaves under salt stress (S) increased significantly with time compared to the control plants (CK). It increased by 102.0%, 87.1%, 61.5%, and 34.5% at 7, 14, 21, and 28 d, respectively. Under different concentrations of NaHS treatment, the proline content showed the trend of first decreased and then increased with increasing NaHS concentration. Compared with single salt stress (S) treatment alone, the proline content decreased significantly with increasing NaHS treatment time. The proline content decreased by 28.0%, 27.3%, 36.0%, and 30.8% compared with S treatment by different periods of the most effective NaHS concentrations of 7 d (0.1 mM NaHS), 14 d (0.25 mM NaHS), 21 d (0.05 mM NaHS), and 28 d (0.05 mM NaHS), respectively. It was shown that exogenous NaHS treatment significantly inhibited the accumulation of proline content in R. soongorica leaves under salt stress.

Figure 2 Effects of exogenous H2 S (NaHS) on the proline content (A), malondialdehyde (MDA) content (B), soluble sugar content (C) and soluble protein content (D) of R. soongorica seedlings under salt stress.

As shown in Fig. 2B, the MDA content in R. soongorica leaves was significantly increased under salt stress treatment compared with CK. It increased by 91.6%, 66.9%, 51.0%, and 73.7% at 7, 14, 21, and 28 d, respectively. Through different concentrations of NaHS treatment, the MDA content showed the trend of first decreased and then increased with increasing NaHS concentration. Compared with S plants, the MDA content decreased significantly with increasing NaHS treatment time. And the NaHS treatment concentration (0.1 mM) was the most significant at 7 d, the MDA content decreased by 37.2% compared with S plants; while the NaHS treatment concentration (0.05 mM) was the most significant at 14, 21, and 28 d, the MDA content decreased by 35.8%, 35.1%, and 34.6% compared with S plants, respectively. It was demonstrated that exogenous NaHS treatment effectively inhibited the accumulation of MDA content in R. soongorica leaves under salt stress.

As shown in Fig. 2C, the soluble sugar (SS) content in R. soongorica leaves gradually decreased under salt stress compared to CK. It decreased by 6.5%, 4.0%, 6.5%, and 6.8% at 7, 14, 21, and 28 d, respectively. Under different concentrations of NaHS treatment, the SS content showed a trend of first increased and then decreased with increasing NaHS concentration. The SS content increased significantly with increasing NaHS treatment time compared with S plants. In particular, the SS content under 0.025 mM NaHS treatment increased by 11.7%, 13.9%, 17.8%, and 18.9% compared with S plants at 7, 14, 21, and 28 d, respectively. It was concluded that exogenous NaHS treatment significantly increased the SS content in R. soongorica leaves under salt stress.

As shown in Fig. 2D, the soluble protein (SP) content in R. soongorica leaves showed a trend of first increased and then decreased under salt stress treatment compared with CK. It increased by 3.3%, 0.7%, 0.1%, and 0.2% at 7, 14, 21, and 28 d, respectively. After different concentrations of NaHS treatment, the SP content showed a trend of first increased and then decreased with increasing NaHS concentration. Compared with S plants, the SP content increased significantly with increasing NaHS treatment time. In particular, compared with S plants, the SP content under the treatment with 0.025 mM NaHS concentration increased by 5.7%, 5.8%, 4.7%, and 3.7% at 7, 14, 21, and 28 d, respectively. It showed that exogenous NaHS treatment effectively regulated the effect of salt stress on the SP content of R. soongorica.

Exogenous H2S affects antioxidant enzymes of R. soongorica
seedlings under salt stress

As shown in Fig. 3A, the SOD activity of R. soongorica leaves was significantly increased under salt stress treatment (S) compared with the control plant (CK), which increased by 24.4%, 20.6%, 24.3%, and 15.0% at 7, 14, 21, and 28 d, respectively. Under different concentrations of NaHS treatment, the SOD activity showed the trend of first increased and then decreased with increasing NaHS concentration. Compared with S plants, the SOD activity increased considerably with increasing NaHS treatment time. And the NaHS treatment (0.025 mM) was the most significant at 7 d, the SOD activity increased by 33.9% compared with S plants; while the NaHS treatment concentration (0.05 mM) was the most significant at 14, 21, and 28 d, and increased by 40.4%, 41.7%, and 47.2% compared with S plants, respectively. It showed that exogenous NaHS effectively increased the SOD activity of R. soongorica leaves under salt stress.

Figure 3 Effects of exogenous H2S (NaHS) on the superoxide dismutase (SOD) activity (A), peroxidase (POD) activity (B), catalase (CAT) activity (C) and superoxide anion (O2−) content (D) of R. soongorica seedlings.

It can be seen from Fig. 3B that the POD activity of R. soongorica leaves under salt stress treatment showed different degrees of increase compared with CK. By foliar spraying of exogenous NaHS, the POD activity showed a trend of first increased and then decreased with increasing NaHS concentration. Compared with S, the POD activity increased significantly with increasing NaHS treatment time. The effect of 0.05 mM NaHS treatment was the most significant at 14 d, when the POD activity increased by 33.8% compared with S plants; the NaHS treatment concentration (0.025 mM) was the most significant at 7, 21, and 28 d, which increased by 41.1%, 53.4%, and 92.5% compared with S plants, respectively. It was demonstrated that exogenous NaHS effectively promoted the POD activity of R. soongorica leaves under salt stress.

As shown in Fig. 3C, the CAT activity was significantly increased under salt stress compared with CK. It increased by 49.3%, 74.0%, 49.7%, and 31.1% at 7, 14, 21, and 28 d, respectively. After different concentrations of NaHS treatment, the CAT activity showed the trend of first increased and then decreased with increasing NaHS concentration. The CAT activity increased significantly with increasing NaHS treatment time compared with S plants. In particular, the NaHS treatment concentration (0.025 mM) was the most significant, and the CAT activity increased by 75.1%, 63.6%, 74.9%, and 75.8% at 7, 14, 21, and 28 d, respectively, compared with S plants. It was demonstrated that exogenous NaHS effectively enhanced the CAT activity of R. soongorica leaves under salt stress.

As shown in Fig. 3D, the superoxide anion (O2−) content of R. soongorica leaves was significantly increased under salt stress compared with CK. It increased by 121.0%, 115.7%, 71.5%, and 64.6% at 7, 14, 21, and 28 d, respectively. Under different concentrations of NaHS treatment, the O2− content showed a trend of first decreased and then increased with increasing NaHS concentration. Compared with S, the O2− content decreased significantly with increasing NaHS treatment time. Especially when the NaHS treatment concentration (0.1 mM) had the most significant effect on alleviating salt damage in R. soongorica under salt stress, which was reduced by 29.0%, 30.9%, 30.5%, and 35.7% at 7, 14, 21, and 28 d, respectively, compared with S. It showed that exogenous NaHS effectively reduced the accumulation of O2− content in R. soongorica leaves under salt stress.

Exogenous H2S affects endogenous H2S content and LCD enzyme activity in R. soongorica seedlings under salt stress

It is shown in Fig. 4A that salt stress treatment (S) significantly increased the endogenous H2S content of R. soongorica leaves compared with the control plant (CK). It increased by 27.8%, 49.0%, 53.9%, and 79.1% at 7, 14, 21, and 28 d, respectively; the LCD activity showed the same trend as the endogenous H2S content (Fig. 4B), and LCD activity increased by 24.0%, 22.7%, 39.1%, and 33.2% at 7, 14, 21, and 28 d, respectively, compared with CK. After different concentrations of NaHS treatment, the endogenous H2S content and the LCD activity gradually increased with increasing NaHS concentration compared with S. The endogenous H2S content and the LCD activity also showed an increasing trend with increasing NaHS treatment time. It showed that the external application of NaHS had a significant promoting effect on the endogenous H2S content and the LCD activity of R. soongorica leaves.

Figure 4 Effects of exogenous H2S (NaHS) on the hydrogen sulfide (H2S) content (A), L-cysteine desulfhydrase (LCD) activity (B) of R. soongorica seedlings under salt stress.

Principal component analysis

The principal component analysis of the 14 indicators was performed using SPSS 26.0 software under salt stress, and the total variance interpretation of the principal components was obtained using the mean of the relevant indicators combined (Table 1).

From Table 1, the cumulative contribution was 95.674% after 7 d of NaHS treatment; and the cumulative contribution was 92. 789% after 14 d of NaHS treatment; and the cumulative contribution was 92.799% after 21 d of NaHS treatment; and the cumulative contribution was 95.595% after 28 d of NaHS treatment. All four eigenvalues of NaHS treatment time were greater than 1, and the cumulative contribution was greater than 90.000%. It showed that the first three principal components of R. soongorica seedlings represented the information of the 14 original salt stress indicators by 95.674%, 92.789%, 92.799%, and 95.595%, respectively.

Based on the principle that the eigenvalues of the principal components of the four NaHS treatment times were >1 and the cumulative variance contribution rate was >80%, the top three principal components were selected as the main evaluation indicators of the alleviating effect of exogenous H2S on R. soongorica seedlings under salt stress (Table 2). By ranking the eigenvectors of the four NaHS treatment times in descending order, the top three and the most frequent cumulative eigenvectors were selected by comprehensive analysis: LRWC (0.940, 0.244, 0.961, 0.880, 0.968), POD (0.911, 0.915, 0.910, 0.957), and proline (0.835, 0.779, 0.902, 0.754) were used as the main evaluation indicators of the alleviating effect of exogenous H2S on R. soongorica seedlings under salt stress.

The comprehensive score calculated using the principal component linear equation (Table 3) showed that the higher the principal component score for the treatment with exogenous H2S treatment, the more significant the effect of this NaHS treatment concentration on the alleviation of R. soongorica under salt stress was indicated. It was found that the most alleviating significant effect of 0.025 mM NaHS treatment was observed at 7, 14, 21, and 28d, which indicated that 0.025 mM NaHS should be the optimal alleviating concentration of exogenous H2S for R. soongorica seedlings under salt stress.

Table 3 Each principal component score and principal composite score of leaves of R. soongorica seedlings under salt stress alleviated by exogenous hydrogen sulfide at different times.

Treatment time (day)	Principal component	Treatment	
		CK	S	S + T1	S + T2	S + T3	S + T4	S + T5	S + T6	S + T7	
7	Principal component 1	−0.067	−5.845	−0.578	1.384	1.876	2.072	1.535	0.323	−0.700	
Principal component 2	−5.996	0.476	1.561	1.803	0.788	−0.152	0.096	0.491	0.933	
Principal component 3	−0.546	−0.398	−1.717	−1.739	−0.585	0.066	0.732	1.751	2.437	
synthesis score	−2.445	−2.334	0.115	1.032	1.012	0.823	0.791	0.583	0.424	
14	Principal component 1	−6.365	0.640	1.371	1.622	0.432	0.432	0.215	0.650	1.003	
Principal component 2	−0.096	−5.455	−1.088	1.591	2.412	1.981	0.981	−0.010	−0.317	
Principal component 3	0.385	0.257	1.440	1.996	0.183	−0.259	−0.735	−1.449	−1.818	
synthesis score	−2.685	−1.858	0.308	1.534	1.156	0.937	0.398	0.111	0.099	
21	Principal component 1	−4.415	−4.098	0.299	2.312	2.242	2.017	1.221	0.459	−0.038	
Principal component 2	−4.448	3.937	1.350	0.325	−1.266	−0.859	−0.112	0.200	0.873	
Principal component 3	0.361	0.475	1.143	1.626	0.627	−0.395	−0.400	−1.582	−1.857	
synthesis score	−3.641	−0.458	0.739	1.362	0.657	0.598	0.494	0.133	0.115	
28	Principal component 1	−4.411	−4.775	0.518	2.568	2.304	2.246	1.125	0.630	−0.205	
Principal component 2	−4.139	3.405	1.548	0.864	−1.156	−0.835	−0.397	0.130	0.580	
Principal component 3	0.662	0.322	1.084	1.703	0.586	−0.197	−0.740	−1.527	−1.893	
synthesis score	−3.614	−1.529	0.870	1.843	0.963	0.950	0.417	0.228	−0.127	
Notes.

Control (CK) and salt-treated (S) plants were sprayed with distilled water, while the S + T1, S + T2, S + T2, S + T3, S + T4, S + T5, S + T6, S + T7 plants were sprayed with 0.01, 0.025, 0.05, 0.1, 0.25, 0.5, 1 mM NaHS, respectively.

Affiliation function analysis

The affiliation function values were calculated based on 14 indicators of R. soongorica seedlings (Table 4). The results showed that the external application of NaHS at a concentration of 0.025 mM had the most alleviating significant effect at 7, 14, 21, and 28 d. The same conclusion as the above comprehensive score of principal components further proved that 0.025 mM NaHS had the most significant alleviating effect on R. soongorica seedlings under salt stress.

Table 4 Affiliation function values of leaves of R. soongorica seedlings under salt stress alleviated by exogenous hydrogen sulfide at different times.

T (day)	TC
(mM)	CAT	H2S	LCD	MDA	POD	SOD	O2−	SP	Pro	SS	GD-AGR	LRWC	SLA	PH-RGR	Mean	RK	
7	CK	0.052e	0.020f	0.033f	0.075 g	0.037 g	0.050e	0.024f	0.071e	0.018e	0.577d	0.634a	0.587a	0.741ab	0.791a	0.265		
S	0.328d	0.184e	0.173e	0.831a	0.528a	0.368d	0.925e	0.380d	0.809d	0.020 g	0.118b	0.127b	0.114e	0.268b	0.370		
S+T1	0.723abc	0.232e	0.218e	0.752a	0.901a	0.803abc	0.799e	0.838abc	0.626abc	0.653c	0.387ab	0.601ab	0.517cd	0.713ab	0.626	6	
S+T2	0.956a	0.420d	0.378d	0.635b	0.985b	0.919a	0.634d	0.935a	0.530a	0.955a	0.398ab	0.697ab	0.714abc	0.798ab	0.711	1	
S+T3	0.805ab	0.471d	0.536d	0.379e	0.897e	0.893ab	0.550d	0.795ab	0.514ab	0.787b	0.516ab	0.806ab	0.893a	0.701ab	0.682	2	
S+T4	0.716abc	0.475d	0.597d	0.242f	0.871f	0.832abc	0.447d	0.743abc	0.370abc	0.738b	0.731a	0.795a	0.669bcd	0.716a	0.639	4	
S+T5	0.658bc	0.661c	0.823c	0.455de	0.835de	0.772bc	0.461c	0.594bc	0.412bc	0.611cd	0.613ab	0.729ab	0.649bcd	0.868ab	0.653	3	
S+T6	0.519cd	0.830b	0.895b	0.515cd	0.789cd	0.787cd	0.547b	0.563cd	0.474cd	0.476e	0.602ab	0.703ab	0.512d	0.601ab	0.630	5	
S+T7	0.481cd	0.971a	0.937a	0.590bc	0.754bc	0.737cd	0.594a	0.490cd	0.593cd	0.211f	0.409ab	0.759ab	0.537cd	0.497ab	0.611	7	
14	CK	0.073e	0.012e	0.042e	0.039d	0.019d	0.088e	0.032e	0.172e	0.043e	0.298e	0.487a	0.561a	0.682ab	0.833a	0.242		
S	0.408d	0.233d	0.164d	0.911a	0.597a	0.339d	0.925d	0.249d	0.817d	0.012f	0.154a	0.161a	0.141d	0.097a	0.372		
S+T1	0.610bcd	0.523c	0.351c	0.673b	0.821b	0.599bcd	0.828c	0.633bcd	0.560bcd	0.636b	0.179a	0.582a	0.407c	0.368a	0.555	7	
S+T2	0.909a	0.667b	0.465b	0.623b	0.867b	0.812a	0.648b	0.873a	0.507a	0.955a	0.346a	0.631a	0.588bc	0.557a	0.675	1	
S+T3	0.741ab	0.653b	0.622b	0.132d	0.949d	0.930ab	0.449b	0.772ab	0.427ab	0.416d	0.385a	0.751a	0.787a	0.460a	0.605	2	
S+T4	0.691abc	0.691b	0.635b	0.411c	0.903c	0.862abc	0.411b	0.654abc	0.414abc	0.407d	0.372a	0.772a	0.841a	0.399a	0.604	3	
S+T5	0.635bcd	0.703b	0.798b	0.417c	0.787c	0.677bcd	0.455b	0.520bcd	0.363bcd	0.476c	0.372a	0.701a	0.579bc	0.326a	0.558	6	
S+T6	0.479cd	0.885a	0.953a	0.619b	0.780b	0.661cd	0.482a	0.468cd	0.425cd	0.450cd	0.346a	0.547a	0.464c	0.303a	0.561	5	
S+T7	0.476cd	0.959a	0.974a	0.718b	0.730b	0.631cd	0.567a	0.426cd	0.603cd	0.352e	0.359a	0.621a	0.489c	0.281a	0.585	4	
21	CK	0.018f	0.004c	0.009c	0.178e	0.017e	0.042f	0.088c	0.149f	0.062f	0.384de	0.949a	0.420a	0.594a	0.711a	0.259		
S	0.301e	0.138c	0.183c	0.960a	0.499a	0.301e	0.979c	0.158e	0.911e	0.034f	0.231a	0.091a	0.137b	0.047a	0.355		
S+T1	0.555cd	0.434b	0.639b	0.681bc	0.851bc	0.656cd	0.810b	0.418cd	0.496cd	0.927a	0.359a	0.549a	0.402ab	0.288a	0.576	4	
S+T2	0.939a	0.611a	0.704a	0.576cd	0.980cd	0.811a	0.708a	0.656a	0.342a	0.927a	0.487a	0.684a	0.597a	0.309a	0.666	1	
S+T3	0.764b	0.616a	0.759a	0.147e	0.919e	0.851b	0.546a	0.597b	0.109b	0.530b	0.513a	0.739a	0.761a	0.316a	0.583	2	
S+T4	0.660bc	0.630a	0.894a	0.434d	0.874d	0.746bc	0.326a	0.578bc	0.241bc	0.451cd	0.436a	0.853a	0.747a	0.276a	0.582	3	
S+T5	0.530d	0.634a	0.915a	0.447d	0.835d	0.693d	0.470a	0.508d	0.181d	0.493bc	0.333a	0.685a	0.527ab	0.297a	0.539	7	
S+T6	0.381e	0.662a	0.956a	0.698bc	0.843bc	0.686e	0.477a	0.245e	0.341e	0.407de	0.359a	0.690a	0.575ab	0.396a	0.551	6	
S+T7	0.342e	0.768a	0.962a	0.762b	0.774b	0.689e	0.530a	0.197e	0.545e	0.364e	0.359a	0.588a	0.490ab	0.415a	0.556	5	
28	CK	0.042e	0.035e	0.011e	0.020f	0.040f	0.023e	0.100e	0.214e	0.221e	0.391 g	0.769a	0.324a	0.591bc	0.926a	0.265		
S	0.226de	0.229d	0.150d	0.971a	0.242a	0.195de	0.888d	0.241de	0.900de	0.023 h	0.128a	0.109a	0.080d	0.164a	0.325		
S+T1	0.403bcd	0.584c	0.602c	0.629c	0.924c	0.735bcd	0.802c	0.525bcd	0.394bcd	0.934b	0.205a	0.586a	0.434c	0.264a	0.573	3	
S+T2	0.814a	0.791b	0.719b	0.562c	0.965c	0.790a	0.571b	0.753a	0.385a	0.985a	0.256a	0.766a	0.763ab	0.248a	0.669	1	
S+T3	0.624ab	0.818b	0.743b	0.196e	0.932e	0.817ab	0.257b	0.564ab	0.085ab	0.887c	0.410a	0.738a	0.782ab	0.293a	0.582	2	
S+T4	0.518bc	0.855ab	0.840ab	0.357d	0.881d	0.749bc	0.171ab	0.537bc	0.086bc	0.839d	0.282a	0.665a	0.957a	0.236a	0.570	4	
S+T5	0.493bc	0.839b	0.847b	0.385d	0.831d	0.698bc	0.283b	0.333bc	0.166bc	0.652e	0.282a	0.628a	0.711abc	0.310a	0.533	5	
S+T6	0.336cd	0.886ab	0.926ab	0.579c	0.842c	0.670cd	0.243ab	0.229cd	0.287cd	0.631e	0.308a	0.620a	0.614bc	0.243a	0.530	6	
S+T7	0.236de	0.958ab	0.956ab	0.752b	0.733b	0.567de	0.324ab	0.240de	0.380de	0.462f	0.231a	0.515a	0.612bc	0.400a	0.526	7	
Notes.

Data are mean (n = 3). Different lowercase letters denote significant differences at the 0.05 probability level according to Duncan test. The T in the table represents treatment time; TC represents the treatment concentration. Mean in the table represents the membership function mean; RK represents the ranking of the mitigating effects of exogenous H2S on R. soongorica seedlings under salt stress.

Exogenous H2S effects on the biomass of R. soongorica seedlings under salt stress

As shown in Table 5, the fresh and dry weights of above-ground parts of R. soongorica seedlings were significantly lower than those of CK under salt stress, but the fresh and dry weights of below-ground parts were higher than those of CK. Similarly, Fig. 5 further demonstrates that salt stress significantly inhibited the accumulation of aboveground biomass of R. soongorica seedlings and increased the level of root biomass of R. soongorica seedlings. The biomass accumulation levels of above-ground and below-ground parts of R. soongorica seedlings treated with different concentrations of exogenous NaHS were higher than those of CK and S. The total biomass of R. soongorica seedlings also showed the same trend of change as that of exogenous NaHS treatment. After 28 d of treatment, the results showed that the effect of exogenous H2S on the biomass accumulation of R. soongorica seedlings under salt stress was 0.025 mM NaHS >0.05 mM NaHS >0.1 mM NaHS >0.25 mM NaHS >0.01 mM NaHS >0.5 mM NaHS >1 mM NaHS in descending order. This is the same conclusion as the above principal component score and affiliation function, which further proved that 0.025 mM NaHS treatment had the most pronounced alleviating effect on R. soongorica seedlings under salt stress.

Table 5 Effect of exogenous H2S on the biomass of R. soongorica seedlings under salt stress.

Treatment (mM)	Above-ground fresh weight	Above-ground
dry weight	Below-ground
fresh weight	Below-ground
dry weight	Whole plant
fresh weight	Whole plant
dry weight	
	g ⋅ plant −1	
CK	4.470 ± 0.709ab	1.289 ± 0.280ab	0.468 ± 0.106b	0.339 ± 0.078a	4.938 ± 0.687b	1.628 ± 0.286ab	
S	4.240 ± 1.004b	1.162 ± 0.377b	0.518 ± 0.220ab	0.369 ± 0.173a	4.758 ± 1.208b	1.531 ± 0.545c	
S+T1	5.043 ± 0.737ab	1.466 ± 0.221ab	0.753 ± 0.161ab	0.485 ± 0.062a	5.796 ± 0.675ab	1.951 ± 0.216abc	
S+T2	6.276 ± 1.061a	1.941 ± 0.356a	0.864 ± 0.120a	0.577 ± 0.036a	7.140 ± 0.996a	2.518 ± 0.326a	
S+T3	6.303 ± 1.661a	1.859 ± 0.488a	0.779 ± 0.117ab	0.560 ± 0.095a	7.082 ± 1.754a	2.419 ± 0.577ab	
S+T4	5.848 ± 0.810ab	1.872 ± 0.283a	0.769 ± 0.127ab	0.512 ± 0.063a	6.617 ± 0.847ab	2.385 ± 0.289abc	
S+T5	5.617 ± 0.516ab	1.578 ± 0.239ab	0.686 ± 0.059ab	0.457 ± 0.070a	6.303 ± 0.571ab	2.035 ± 0.307abc	
S+T6	4.902 ± 1.375ab	1.409 ± 0.553ab	0.746 ± 0.414ab	0.506 ± 0.281a	5.647 ± 1.711ab	1.915 ± 0.802abc	
S+T7	4.899 ± 0.638ab	1.391 ± 0.259ab	0.692 ± 0.190ab	0.460 ± 0.130a	5.590 ± 0.690ab	1.851 ± 0.319abc	
Notes.

Data were presented as average ± SE ( n = 3). Different lowercase letters denote significant differences at the 0.05 probability level according to Duncan test.

Figure 5 A representative figure of the phenotypic differences of R. soongorica seedlings under different treatments.

Discussion

Currently, salt stress is one of the most detrimental factors among abiotic stresses, affecting various physiological metabolic processes in plants and causing severe growth dysfunction, such as ion homeostasis, protein synthesis, and energy metabolism (Genisel, Erdal & Kizilkaya, 2015). Although plants have evolved several defense strategies to resist the effects of salt stress, such as maintaining ion homeostasis in plant cells and increasing the levels of osmoregulatory and antioxidant enzyme systems (Yadav et al., 2020). However, if the plant’s own defense strategies are unable to cope with the effects of salt damage as the degree and duration of salt stress changes, the plant will be unable to grow. Therefore, different induction methods can be applied to enhance salt tolerance in plants, one of which is to mitigate the adverse effects of salt stress on plants by exogenous application of various substances (Kumari et al., 2021; Turk et al., 2014; Zulfiqar et al., 2022). Many studies have shown that exogenous application of some endogenous hormones can effectively alleviate the adverse effects of salt stress on plants (Chen, Cao & Niu, 2021; Kurepin et al., 2015; Li et al., 2022). H2S has received increasing attention as a potential bioregulator (Lai et al., 2014).

Plants respond to different abiotic stresses through the exogenous use of H2S (Li et al., 2012). The endogenous H2S content of the same plant based on abiotic stress increases by an average of 2.0−2.5 times compared to plants under normal conditions (Banerjee, Tripathi & Roychoudhury, 2018). After salt stress treatment, the LCD activity of Arabidopsis was significantly increased, and the level of endogenous H2S was increased by 1.2–3 times (Shi et al., 2015). In addition, the total LCD activity and the production of endogenous H2S in alfalfa seedlings increased as the concentration of NaCl increased (Lai et al., 2014). In this study, salt stress treatment significantly increased the endogenous H2S content and LCD activity of R. soongorica seedlings, and the endogenous H2S content and LCD activity increased significantly with increasing treatment time under the continuous application of exogenous NaHS. This is consistent with the results of related studies that the endogenous H2S content and LCD activity were effectively increased in cucumber by application of exogenous NaHS under nitrate stress (Qi et al., 2019). Similarly, exogenous NaHS treatment induced an increase in endogenous H2S levels and LCD enzyme activity in tomato plants after nitrate treatment (Guo et al., 2018). The endogenous H2S content and LCD activity of cucumber seedlings were also significantly increased by NaHS treatment under NaCl stress (Jiang et al., 2019). The main reason may be the ability of plants to actively synthesize endogenous H2S when they are under biotic or abiotic stress and is also closely related to plant stress tolerance (Hancock & Whiteman, 2014; Yamasaki & Cohen, 2016). Besides abiotic and biotic stressors, plants can also activate H2S signaling in cells by exogenous application of NaHS or by upregulating the expression of genes involved in H2S biosynthesis (Li, Min & Zhou, 2016). For example, 0.1−0.3 mM NaHS treatment promoted the growth of Prunus persica seedlings compared with the control, and exogenous H2S treatment induced the expression of the genes encoding H2S synthases PpDCD and PpLCD in the roots and leaves of Prunus persica seedlings and increased the endogenous H2S content in the roots of seedlings (Gao et al., 2019).

When plants are exposed to salt stress, the main effect on plants is growth inhibition (Gao et al., 2021). Related studies have shown that the reduction in plant growth under salt stress is mainly due to water deficiency, nutrient imbalance and specific toxicity (Chen, Cao & Niu, 2021; Hamid, Khalil ur Rehman & Ashraf, 2010). This is because salt stress can severely affect the plant’s ability to absorb water, leading to a physiological water deficit that reduces the water potential and osmotic potential of plant cells, resulting in a gradual decrease in leaf RWC (James et al., 2002; Munns, 2002). For example, salt stress significantly inhibits the growth of maize seedlings (Erdal, 2012). At the germination stage, salt stress also alters the main growth parameters of Apocynum venetum L. seedlings, causing leaf deformation and reduced plant height of seedlings (Xu et al., 2020). Furthermore, when NaCl concentrations reached 200 mM and above, hawthorn plant growth showed concentration-dependent inhibition, resulting in shortened plant height and reduced leaf RWC (Gao et al., 2021). In this study, the growth of height and ground diameter of R. soongorica seedlings was significantly inhibited, and the LRWC and SLA of R. soongorica were reduced when they were subjected to salt stress. As the duration of salt stress increased, the growth inhibition of R. soongorica seedlings was further aggravated. When exogenous NaHS was added, the growth rate of R. soongorica plant height and ground diameter was significantly improved compared with the salt control (S), and the normal growth of R. soongorica LRWC and SLA was maintained. And it could help them to recover to the control level, effectively alleviating the growth inhibition of R. soongorica seedlings by salt stress. This is consistent with previous findings that the growth of wheat seedlings was significantly inhibited by 100 mM NaCl treatment, and the salt-induced growth inhibition of wheat seedlings was partially reversed after pretreatment with 0.05 mM NaHS (Deng et al., 2015). The plant height, ground diameter, leaf area, and LRWC content of tomato were decreased with the increase of salt stress under salt stress. H2S treatment alleviated the negative effects of salt stress (Yildirim et al., 2023). Meanwhile, excess nitrate inhibited the growth of tomato plants, and the growth inhibition of tomato plants was alleviated by exogenous NaHS treatment (Guo et al., 2018). In addition, salt stress decreased the LRWC of rice plants, and the LRWC of rice was increased by 19% by NaHS treatment compared with the plants under salt stress alone, which significantly suppressed the decrease in LRWC of rice plants under salt stress (Mostofa et al., 2015). The main reason may be that plants under salt stress maintain the nutrients necessary for growth and development by regulating osmotic factors, and increase the cellular osmotic pressure by increasing the concentration of cytoplasmic solutes to reduce the water potential and decrease cellular water loss. And salt-stressed plants can alleviate salt damage at the expense of growth and development. In this study, salt stress significantly inhibited the biomass accumulation of R. soongorica seedlings, and the level of biomass accumulation of R. soongorica was higher than both salt stress treatment and CK by exogenous NaHS treatment, which effectively promoted the growth and development of R. soongorica seedlings. This is consistent with the results of a related study, in which the fresh and dry weights of above-ground and below-ground parts of Avena nuda seedlings under salt stress were significantly lower than those of CK. After 50 µM NaHS treatment, the fresh and dry weights of above-ground and below-ground parts of Avena nuda seedlings increased, which alleviated the inhibitory effect of salt stress on the growth of Avena nuda seedlings to some extent (Liu et al., 2021). Meanwhile, the growth of tomato seedlings was severely inhibited after 1 d of 100 mM NO3− stress treatment compared with the control plants, and the height, root length, and fresh and dry weights of above-ground and below-ground parts of tomato seedlings were significantly reduced. The growth inhibition of tomato seedlings was significantly alleviated by exogenous application of 100 µM NaHS, and their fresh weight, dry weight, plant height, and root length were increased (Jing et al., 2015). In addition, the fresh dry weight and root fresh dry weight of tomato seedlings decreased with increasing salt stress, and the negative effects of salt on these parameters were alleviated by NaHS treatment (Yildirim et al., 2023).

Since soluble sugars and soluble proteins are essential energy sources and important osmoregulatory substances for plant growth, their increased and accumulated levels can effectively improve the water-holding capacity of plant cells when plants are exposed to abiotic stress (Liu et al., 2019; Shi, Ye & Chan, 2013). Combined with the present study, it was found that the soluble sugar content of R. soongorica leaves significantly decreased under salt stress, while the soluble protein content gradually increased. The reason may be that different plants selectively synthesize different organic solutes when they were subjected to abiotic stress. And the application of exogenous NaHS significantly increased the content of soluble sugar and soluble protein, and played a role in protecting the vital substances and biofilm of plant cells. This is consistent with the results of a related study, in which the application of exogenous NaHS under NaCl conditions significantly increased osmotic substances such as soluble sugars in bermudagrass, positively regulated the accumulation of osmotic fluid, and effectively enhanced the stress tolerance of the plant (Shi, Ye & Chan, 2013). Wheat showed a trend toward increased soluble protein content after NaHS pretreatment under salt stress and inhibited the accumulation of ROS, which contributed to the protection against oxidative damage (Mostofa et al., 2015). Furthermore, H2S induced an increase in non-enzymatic antioxidants in Kandelia obovata plants during adaptation to hypersaline environments, which prevented cell damage under high salinity. In addition, proteomic analysis of the regulatory effects of H2S on salt tolerance in mangrove plants revealed that heat shock proteins (HSPs) and chaperone protein family (Cpn) proteins are involved in protein synthesis, and this process is positively regulated by the addition of sodium sulfide (Liu et al., 2019). HSP and Cpn are molecular chaperones involved in protein transport, protein folding and protein assembly processes (Omar et al., 2011). Furthermore, it was shown that exogenous NaHS alleviated the growth stress of R. soongorica seedlings under salt stress by participating in osmoregulation.

Under normal growth conditions, the antioxidant enzyme system of plants has reached a relative balance between ROS production and ROS elimination, maintaining normal ROS levels. Under salt stress, plants will destroy the original ROS homeostasis due to ion toxicity, while inducing plasma membrane instability and reduced cell expansion pressure (Kumar et al., 2016; Kumari, Gupta & Yadav, 2021; Mittler et al., 2011). The end product of salt stress-induced membrane lipid peroxidation is MDA (Li et al., 2012). In the present study, NaCl treatment gradually induced a burst of proline, O2−, and MDA in R. soongorica seedlings, indicating that high levels of reactive oxygen species were generated in R. soongorica under salt stress, leading to membrane lipid peroxidation damage. The levels of proline, O2−, and MDA were significantly reduced by exogenous application of different concentrations of NaHS. This is consistent with previous findings that NaHS-treated Arabidopsis showed lower levels of O2− (Shi et al., 2015). In addition, the application of exogenous NaHS significantly reduced the O2− and MDA content of pea seedlings under arsenate induction (Singh et al., 2015). The proline content of rice plants significantly increased under salt stress, and NaHS pretreatment effectively inhibited the increase of proline content in plant leaves (Mostofa et al., 2015). Meanwhile, under PEG, chromium (Cr) and other osmotic stresses, the increase in lipoxygenase activity and MDA induced by PEG and Cr during wheat seed germination could be reduced by NaHS pretreatment (Zhang et al., 2010). And with the outbreak of ROS, plants have evolved a complex ROS detoxification system that includes enzymatic antioxidant enzymes (SOD, CAT, POD) and non-enzymatic antioxidant molecules (Apel & Hirt, 2004). In this study, the activities of SOD, POD, and CAT of R. soongorica seedlings increased rapidly under salt stress, but gradually decreased with the prolongation of the treatment time. When different concentrations of NaHS were applied, the activities of SOD, POD, and CAT were significantly enhanced. These results were consistent with those of previous studies. Under salt stress, the transcript levels of several antioxidant genes such as Cu/Zn SOD were down-regulated in alfalfa seeds, and the total activities of SOD, POD, and CAT also showed a downward trend, but NaHS treatment significantly prevented the salt-induced decrease in the activities of these enzymes (Wang et al., 2012). The activities of SOD, POD, and CAT in Arabidopsis were significantly increased by NaHS treatment under salt stress (Shi et al., 2015). The main reason may be that the H2S synthesis pathway of R. soongorica seedlings was greatly activated under salt stress, and the endogenous H2S level was significantly increased. Meanwhile, with the production of endogenous H2S and the application of exogenous H2S, on the one hand, the expression of multiple abiotic stress-related genes was activated and also regulated the accumulation of related osmotic substances; on the other hand, the changes in the activities of antioxidant enzymes under salt stress and improved the tolerance of R. soongorica to salt stress. This is further supported by the H2S-mediated stress response model of Arabidopsis thaliana (Shi et al., 2015). It is shown that exogenous NaHS can improve the salt tolerance of R. soongorica and alleviate the oxidative damage to cells by activating the ROS detoxification system and rebuilding the redox homeostasis in vivo.

Plant stress tolerance is a complex process in which multiple regulatory mechanisms cooperate with each other and is comprehensively affected by factors such as environment, time, and other factors. Compared with other experiments, this study has a relatively long-time span in the treatment of plants, which can better reflect the mitigating effect of exogenous NaHS treatment on the growth and development of R. soongorica under salt stress in different time periods, and obtain relatively good research results. Principal component analysis was used to simplify multiple indicators into a few indicators to interpret the information contained in the original indicators for a comprehensive evaluation, making it more objective and reliable (Jolliffe & Cadima, 2016). In this study, the LRWC, POD, and proline were derived from the principal component analysis as the main indicators to evaluate the mitigating effect of exogenous NaHS on the growth and development of R. soongorica under salt stress. Studies have shown that LRWC and proline content are often considered as important indicators for evaluating salt tolerance in plants (El-Shamy et al., 2022). Accumulated proline can reflect the strength of plants under abiotic stress (Mostofa et al., 2015). Although proline accumulation is not always associated with stress tolerance, the turnover of proline metabolism is required for stress tolerance in plants (Kavi Kishor & Sreenivasulu, 2014). Under osmotic stress, the accumulation of solutes such as proline is a common response of plants to maintain water status (Hayat et al., 2012). The proline accumulation in roses was negatively correlated with RWC under salt stress (Chang et al., 2014). In addition, the LRWC and proline were used as key physiological indicators in a study to evaluate the mitigation mechanisms in rice under salt stress (Polash et al., 2018). In evaluating the extent to which morphological and physiological differences among ponderosa pine populations affect the timing of mortality, RWC was selected as one of the most accurate predictors of drought mortality risk (Sapes & Sala, 2021). Another study of heat tolerance in maize found that NaHS pretreatment could improve heat tolerance and may be related to proline (Li, Ding & Du, 2013).

Peroxidases, a large family of enzymes widely distributed in organisms, function to catalyze redox reactions between hydrogen peroxide as an electron acceptor and a variety of electron donors (Passardi et al., 2005). Peroxidases can form a physical barrier in response to various stimuli by catalyzing the cross-linking of cell wall compounds (Passardi, Penel & Dunand, 2004). Plants exposed to acute stress are known to increase their overall peroxidase activity, which play a major role in isolating or eliminating foreign substances (Passardi et al., 2005). In a study on the recombinant expression and identification of lemon peroxidase, the purified peroxidase was found to be salt-resistant by biochemical characterization. Functional annotation of the protein and transcript analysis indicated that lemon peroxidase may be involved in plant defense responses (Pandey et al., 2021). Characteristic analysis of the cassava POD gene family revealed that the cassava POD gene is involved in response to various stresses or related signals and is a potential target gene for genetic improvement of resistance in cassava crop (Wu et al., 2019). And peroxidase has also become a model enzyme for studying the molecular mechanism of vesicle transport accompanied by glycosylation (Yoshida et al., 2003). While among others, peroxidase has been shown to be quite sensitive to air pollution and is one of the most sensitive methods for assessing the biological effects of pollution. (Moraes et al., 2002; Passardi et al., 2005). The detection of peroxidase activity can also be readily detected throughout the life cycle of various plants in a wide range of physiological processes (Passardi et al., 2005).

Finally, the three evaluation indicators selected in this study cover three major aspects of plant phenotype, antioxidant enzymes, and osmoregulatory substances, which can comprehensively elaborate the regulatory role of H2S to participate in the whole process of R. soongorica growth and development and produced good effects. At the same time, the combined with the comprehensive score of principal components and membership function analysis revealed that the best alleviating effect of leaf spraying 0.025 mM NaHS was found in the regulation of physiological metabolism of R. soongorica under salt stress by different NaHS concentration treatments. In summary, this study has done related research on the physiological and biochemical regulation of exogenous H2S on R. soongorica under salt stress, and analyzed the alleviating effect of exogenous H2S on R. soongorica under salt stress from different levels, but still remained at a shallow research level. It is necessary to further explore the molecular mechanism of exogenous H2S regulating R. soongorica under salt stress, and more work is needed to identify the key genes and proteins and their functions in the regulation of salt stress tolerance by exogenous H2S.

Conclusions

It was found that the exogenous application of different concentrations of NaHS could improve the ROS scavenging ability under salt stress by enhancing the antioxidant capacity of R. soongorica seedlings. Furthermore, exogenous H2S treatment could alleviate osmotic stress by promoting the accumulation of osmoregulatory substances such as soluble sugars and proteins. Meanwhile, exogenous H2S can promote the growth and development of R. soongorica seedlings under salt stress. In this experiment, the principal component analysis and the affiliation function method were used to analyze that 0.025 mM NaHS treatment was most effective in promoting the growth and salt tolerance of R. soongorica seedlings, and LRWC, POD, and proline were screened as the main evaluation indicators of the alleviating effect of exogenous H2S on R. soongorica seedlings under salt stress. This study provides an important theoretical basis for understanding the salt tolerance mechanism of R. soongorica and for cultivating high-quality germplasm resources of R. soongorica. Nevertheless, the molecular mechanism of exogenous H2S in regulating R. soongorica seedlings under salt stress needs further investigation.

Supplemental Information

Supplemental Information 1 Data on the regulatory effect of exogenous hydrogen sulfide on salt-stressed Reaumuria soongorica seedlings

Click here for additional data file.

The authors are grateful to the anonymous reviewers for their valuable comments and suggestions.

Additional Information and Declarations

Competing Interests

Author Contributions

Data Availability

The authors declare there are no competing interests.

Hanghang Liu conceived and designed the experiments, analyzed the data, prepared figures and/or tables, and approved the final draft.

Peifang Chong conceived and designed the experiments, performed the experiments, authored or reviewed drafts of the article, and approved the final draft.

Zehua Liu performed the experiments, authored or reviewed drafts of the article, and approved the final draft.

Xinguang Bao performed the experiments, authored or reviewed drafts of the article, and approved the final draft.

Bingbing Tan performed the experiments, authored or reviewed drafts of the article, and approved the final draft.

The following information was supplied regarding data availability:

Raw measurements are available as a Supplementary File.

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
