# Peer review of "Exogenous hydrogen sulfide improves salt stress tolerance of Reaumuria soongorica seedlings by regulating active oxygen metabolism"

_PeerJ, doi:10.7717/peerj.15881_

## Round 0.1 · original submission · Major Revisions

The paper is evaluated by two independent experts in the field. Both reviewers find the work interesting but raised several issues. In particular, reviewer 2 has valid concerns and I largely agree with the reviewer 2. Based on both reviewers' reports, I have decided that your paper may be considered for publication after revision(s).

Reviewer 1 ·

Basic reporting

No comments

Experimental design

No comments

Validity of the findings

No comments

Additional comments

The manuscript entitled, “Exogenous hydrogen sulfide improves salt stress tolerance of Reaumuria soongorica seedlings by regulating active oxygen metabolism" by Liu et al., evaluated the effects of various concentrations of exogenous Sodium hydrosulfide on osmotic substance content and oxidoreductase activity of R. soongorica seedling under salt stress. The exogenous application of different concentrations of NaHS reduced the content of O2-, proline and MDA, increased the activity of antioxidant enzymes and the content of osmoregulators (soluble sugars and soluble proteins). While the LCD enzyme activity and the content of endogenous H2S were further increased with the continuous application of exogenous H2S, the inhibitory effects of salt stress on the growth rate of plant height and ground diameter, the LRWC and SLA were effectively alleviated. Authors examined the mechanism of exogenous Sodium hydrosulfide mitigates impairment of R. soongorica seedling growth under salt stress. They also screened the optimal exogenous Sodium hydrosulfide concentration for ameliorating salt stress.

While this is a well-written, important, and timely article, there are some changes this reviewer would make to make for a more complete examination of the subject:

Comments:

1 The English of manuscript can be polished (minor), there are few typological mistakes.
2, All figures quality may be improved (high resolution).
3, At least one illustrative figure may be provided as to highlight the summary of this study.
4, The authors should cross-check all abbreviations in the manuscript. Initially, define in full name followed by abbreviation.
5, Authors should include future prospective for this area of study.
6, Authors should add few citations to their manuscript such as PMID: 36012729; PMID: 36160964; PMID: 26008650; PMID: 33288031; PMID: 34866296.
7, Authors should discuss the limitations of this study.

Reviewer 2 ·

Basic reporting

The authors claimed that the exogenous foliar application of hydrogen sulfide on Reaumuria soongorica seedlings enhanced salt stress tolerance.

1. What the manuscript fails to adequately address is the novelty; except for the use of principal component analysis, the results presented here are very similar to previous reports, for example, Bermudagrass (Shi et al. 2013).
2. In general, The manuscript contains several grammatical errors that need to be corrected. (Example, Line 30.. replace “by” with “Under”, Line 31 add “ an” appropriate.. line 35 resources “for”… etc).
3. Abstract is too lengthy and need to be concise.

Experimental design

4. The study uses standard methods for different biochemical estimations. However, if the effect of different concentrations of H2S application on growth is considered, it should be addressed in the biomass too. Moreover, since the availability of transcriptome data, the relevance of the activities of SOD, POD, CAT, and MDA with the modulation of corresponding gene expression is worth including.
5. A representative figure of the phenotypic differences of seedlings under pre-treatment with NaHS and treatment with NaHS under salt stress should be presented.

Validity of the findings

6. The manuscript contains a number of small writing errors that need to be corrected.
7. Even though the role of H2S under different stress conditions are well established in different plant species, the effects of H2S on R. soongorica seedlings under salt stress and the underlying molecular mechanism are still unclear. Since R. soongorica is an extreme xerophyte shrub that can tolerate salinity well, this study has broad relevance for the salinity stress research community.
8. The descriptions of the plant, the unique salt gland structure of R. soongorica, and the role of endogenous H2S production under salt stress conditions are too brief to give readers an adequate understanding of their purpose and function.
9. If the authors have data on the biomass of the seedlings, that would be valuable to include.
10. Figure 1A. The figure may be represented as a bar diagram.
11. I think the discussion could include more analysis referring to recent research progress.

Additional comments

N/A

---

## Round 0.2 · Major Revisions

One of the reviewers is not convinced with the revision and I largely agree with the reviewer. Therefore the manuscript needs further revision.

Reviewer 1 has requested that you cite specific references. You are welcome to add it/them if you believe they are relevant. However, you are not required to include these citations, and if you do not include them, this will not influence my decision.

Reviewer 1 ·

Basic reporting

no comments

Experimental design

no comments

Validity of the findings

no comments

Additional comments

Exogenous hydrogen sulfide improves salt stress tolerance of Reaumuria soongorica seedlings by regulating active oxygen metabolism

Major points for improvement-

What do you mean by “a novel gas signaling molecule”

Have you carried out standardization of R. soongorica under varying NaCl stress. T
How 300mM was selected....

Halotolerant organisms are in use nowdays as a sustainable alternative. Use this new study in discussion
“Thermotolerant and halotolerant Streptomyces sp. isolated from Ajuga parviflora having biocontrol activity against Pseudomonas syringae and Xanthomonas campestris acts as a …
Physiological and Molecular Plant Pathology, 102059”

In ROS related lines of text, add couple of lines and cite these-
Thioredoxins as Molecular Players in Plants, Pests, and Pathogens, Plant-Pest Interactions: From Molecular Mechanisms to Chemical Ecology, 107-125
In relation to this “Plants have evolved a wide array of non-enzymatic and enzymatic defense mechanisms”
Comparative structural modeling of a monothiol GRX from chickpea: Insight in iron–sulfur cluster assembly, International journal of biological macromolecules 51 (3), 266-273
Authors have used Hydrogen sulphide as a priming agent, If yes, then a line related to exogenous priming must be added
See, this biopriming and using microbial priming in soil related papers
Influence of different types of explants in chickpea regeneration using thidiazuron seed-priming
Journal of Plant Research, 1-6
Multitrait Pseudomonas sp. isolated from the rhizosphere of Bergenia ciliata acts as a growth-promoting bioinoculant for plants, Front. Sustain. Food Syst. 7 (1097587)


Section” Exogenous H2S affects ROS levels of R. soongorica seedlings under salt stress”. What may be the putative mechanism?
The levels of proline, were significantly reduced by exogenous application of NaHS?Why
There are redundancy in the paper, kindly avoid redundancy if anything is in table, figure then less must be written in text.
In conclusion, write sustainable approaches importance as it is the basis of your study where no any genetic engineering work has been done and still tolerance is achieved.
See some citations and it will be helpful
There are some language, typographical problems and must be removed.

Furthermore, as earlier suggested authors should add few citations to their manuscript such as PMID: 36012729; PMID: 36160964; PMID: 26008650; PMID: 33288031; PMID: 34866296. That would give comprehensive view to their study.

Reviewer 2 ·

Basic reporting

N/A

Experimental design

N/A

Validity of the findings

N/A

Additional comments

N/A

---

## Round 0.3 · accepted · Accept

I appreciate the authors effort in revising the manuscript satisfactorily.

Reviewer 1 ·

Basic reporting

no comment

Experimental design

no comment

Validity of the findings

no comment